# Designing school reopening in the COVID-19 pre-vaccination period in Bogotá, Colombia: A modeling study

Guido España[1]*, Zulma M. Cucunubá[2,3], Hernando Diaz[4], Sean Cavany[1], Nelson Castañeda[5], Laura Rodriguez[6]

1 Department of Biological Sciences and Eck Institute for Global Health, University of Notre Dame, Notre Dame, Indiana, United States of America, 2 MRC Centre for Global Infectious Disease Analysis, J-IDA, Imperial College London, London, United Kingdom, 3 Departamento de Epidemiología Clínica y Bioestadística, Pontificia Universidad Javeriana, Bogotá, Colombia, 4 Universidad Nacional de Colombia, Bogotá, Colombia, 5 Escuela Tecnológica Instituto Técnico Central, Bogotá, Colombia, 6 GCFEP-Universidad del Tolima, Ibagué, Colombia

* guido.espana@nd.edu

**Data Availability Statement:** The model code is available at: https://github.com/confunguido/FRED The code used to simulate the experiments and to

## Abstract

The COVID-19 pandemic has affected millions of people around the world. In Colombia, 1.65 million cases and 43,495 deaths were reported in 2020. Schools were closed in many places around the world to slow down the spread of SARS-CoV-2. In Bogotá, Colombia, most of the public schools were closed from March 2020 until the end of the year. School closures can exacerbate poverty, particularly in low- and middle-income countries. To reconcile these two priorities in health and fighting poverty, we estimated the impact of school reopening for in-person instruction in 2021. We used an agent-based model of SARS-CoV-2 transmission calibrated to the daily number of deaths. The model includes schools that represent private and public schools in terms of age, enrollment, location, and size. We simulated school reopening at different capacities, assuming a high level of face-mask use, and evaluated the impact on the number of deaths in the city. We also evaluated the impact of reopening schools based on grade and multidimensional poverty index. We found that school at 35% capacity, assuming face-mask adherence at 75% in>8 years of age, had a small impact on the number of deaths reported in the city during a third wave. The increase in deaths was smallest when only pre-kinder was opened, and largest when secondary school was opened. At larger capacities, the impact on the number of deaths of opening pre-kinder was below 10%. In contrast, reopening other grades above 50% capacity substantially increased the number of deaths. Reopening schools based on their multidimensional poverty index resulted in a similar impact, irrespective of the level of poverty of the schools that were reopened. The impact of schools reopening was lower for pre-kinder grades and the magnitude of additional deaths associated with school reopening can be minimized by adjusting capacity in older grades.

generate figures in the manuscript can be found at:
https://github.com/confunguido/manuscript_bogota_school_reopening.

**Funding:** GE received funding from an NSF RAPID grant (DEB 2027718). HD received partial funding from the National University of Colombia (Universidad Nacional de Colombia (HERMES 50419)). ZMC receives funding from the Medical Research Council (MR/R024855/1). The funders had no role in study design, data collection and analysis, decision to publish, or preparation of the manuscript.

**Competing interests:** I have read the journal's policy and the authors of this manuscript have the following competing interests: ZMC holds an honorary role (without payment) in the Scientific Advisory Group on epidemiological modelling at Secretaría de Salud in Bogota.

# Introduction

The COVID-19 pandemic has caused many deaths around the world and in Colombia. As of January 2021, more than 53 thousand COVID-19 deaths had been reported in Colombia. In Bogotá alone, more than 12 thousand people died in the same period. Several interventions have been put in place to curb the spread of SARS-CoV-2, such as city-wide and partial lockdowns, mandatory use of face masks, contact tracing, and school closures [1]. Although interventions such as lockdowns can lead to drastic, albeit temporary, reductions in COVID-19 incidence, they also have negative impacts in society, especially in vulnerable communities [2, 3]. In general, these closures disproportionately affect populations in lower socio-economic groups [4–6]. For instance, the ability of children to learn can be affected by school closures, since virtual learning requires guidance from parents. School closures can also increase the risk of harm by being out of school, such as domestic violence [7].

Schools are important for transmission of respiratory pathogens [8, 9], but the magnitude of their contribution to SARS-CoV-2 transmission is still unclear. School-aged children who are infected with SARS-CoV-2 have a lower chance of developing symptoms of COVID-19, and those who develop symptoms mostly experience milder clinical outcomes [10–12]. However, even if the risk of severe outcomes in children is lower, schools remain a potential source of transmission, which could have downstream effects in the community. In this regard, some limited evidence suggests that children under 10 years of age may be less susceptible to infection [13–15], but the evidence is not conclusive [13]. On the other hand, some studies suggest that children in secondary school could play a much more important role in transmission [16]. In fact, some studies suggest that secondary schools could have contributed to the spread of SARS-CoV-2 earlier in the pandemic [14, 17, 18].

School reopening in the second semester of 2020 in various countries provided additional information about the impact of schools on COVID-19 dynamics. Some studies suggest that outbreaks within schools can be controlled, while others have shown some outbreaks linked to schools. In Israel, large outbreaks were reported just 10 days after reopening [19]. In contrast, school reopening in England during summer 2020 showed that outbreaks in schools were uncommon and strongly related to the local incidence [20]. Similarly, the European CDC concluded that community transmission affected in-school incidence, but that school staff did not have a higher risk than other occupations [21]. In the United States, a study of 11 schools in North Carolina concluded through contact tracing that only 32 infections were acquired within schools and that adults were not infected by children [22]. A study in Mississippi showed evidence that attending in-person school or child care was not associated with increased risk of testing positive for SARS-CoV-2, but participating in social gatherings was [23]. However, given the lower probability of developing symptoms in children, it is difficult to assess the contribution of school reopening in specific communities. Hence, the risk of reopening schools should be evaluated in the local context.

Models are an important tool to understand the dynamics of infectious diseases and to plan public health interventions. Mathematical models have been used to estimate the potential burden of COVID-19 around the world [24–26]. The transmission of SARS-CoV-2 can be heterogeneous across demographic and geographic characteristics of the population. For instance, early non-pharmaceutical interventions implemented to curb the impact of COVID-19 required the ability of people to stay at home for a prolonged period, creating heterogeneous contact patterns in the population, with a potentially higher contact rate in lower income settings. In contrast to compartmental models, agent-based models are capable of incorporating different levels of heterogeneity in transmission due to various factors, such as contact rates or adherence to public health interventions. For instance, in Chile, a stochastic mechanistic

model has shown that early lockdowns were effective to reduce the impact of COVID-19 in Santiago de Chile, but they disproportionately benefited wealthier communities while penalizing vulnerable populations [2]. Within the context of school reopenings, various models suggest that the risk of reopening schools could be minimized with the use of interventions such as reduced class size, face-mask wearing, contact reduction by clustering students [27–31]. Importantly, these models agree that the risk of reopening is higher for older ages. In this study, we evaluate the impact of school reopening in the local context of Bogotá, Colombia, with the use of a stochastic agent-based model of COVID-19 dynamics calibrated to demographic, geographical, education characteristics, and epidemiological information of the city. We evaluated the impact of opening schools by grade and by the school-specific multidimensional poverty index, as well as of opening at different capacities on different dates.

## Results

Our model captured the daily trends of deaths reported in Bogotá over time, space, and age (Fig 1A–1D, and Fig L in S1 Text). To capture the increase in transmission from December to January, an increase in community contacts of 61% was required (95% CI: 60%-65%) in addition to the increased mobility (Fig B in S1 Text). The model slightly underestimated the magnitude of the second peak in January. Compared to 127 reported deaths, the model estimated 103 (95% CrI: 74–145). In addition, the model captured trends of cumulative and age-stratified deaths by localities (Figs C, D in S1 Text). Although the model reproduced the dynamics in

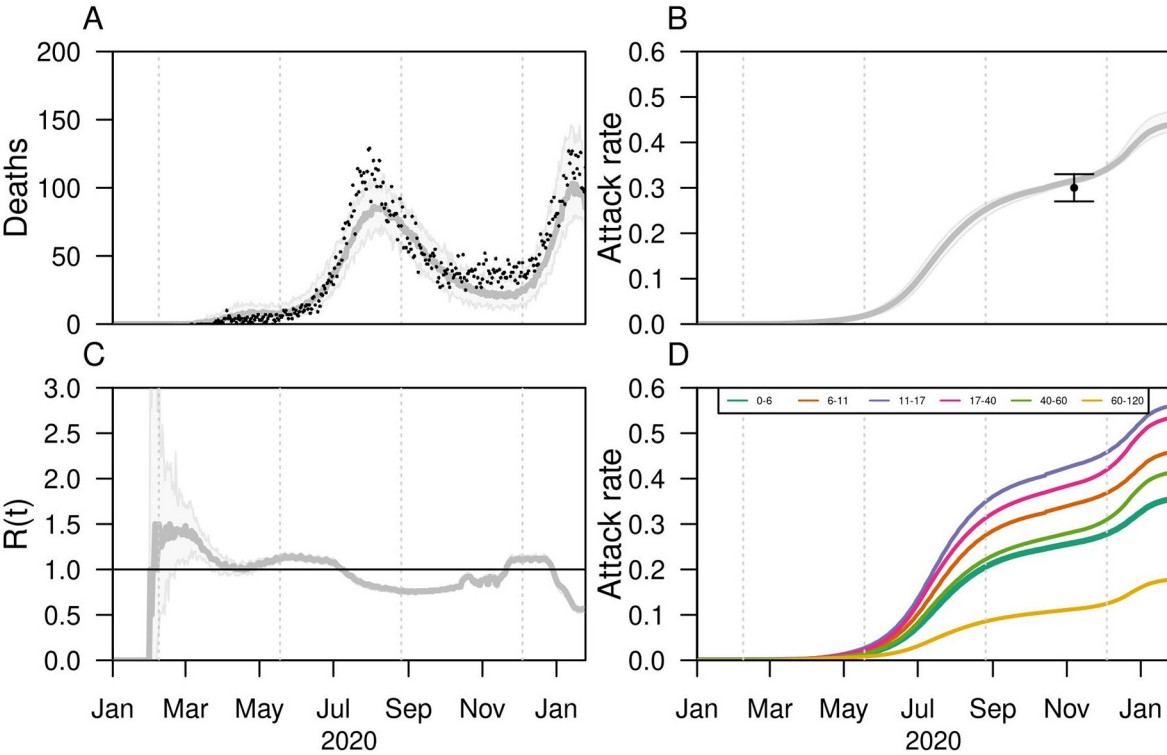

**Fig 1. Model fit to data in Bogotá, Colombia.** Assumption of lower (50%) susceptibility in <10 years. A) Model fit to daily incidence of deaths. Black dots show the official data, and gray lines show the median estimate of the model with the 95% CrI represented by gray-shaded curves. B) Model estimates of attack rate in time represented by gray line (median) and shaded area (95% CrI). The point and arrows show the median estimates and CI of official serological study in Bogotá. C) Estimated reproduction number in time. D) Estimated attack rate in time for different age groups.

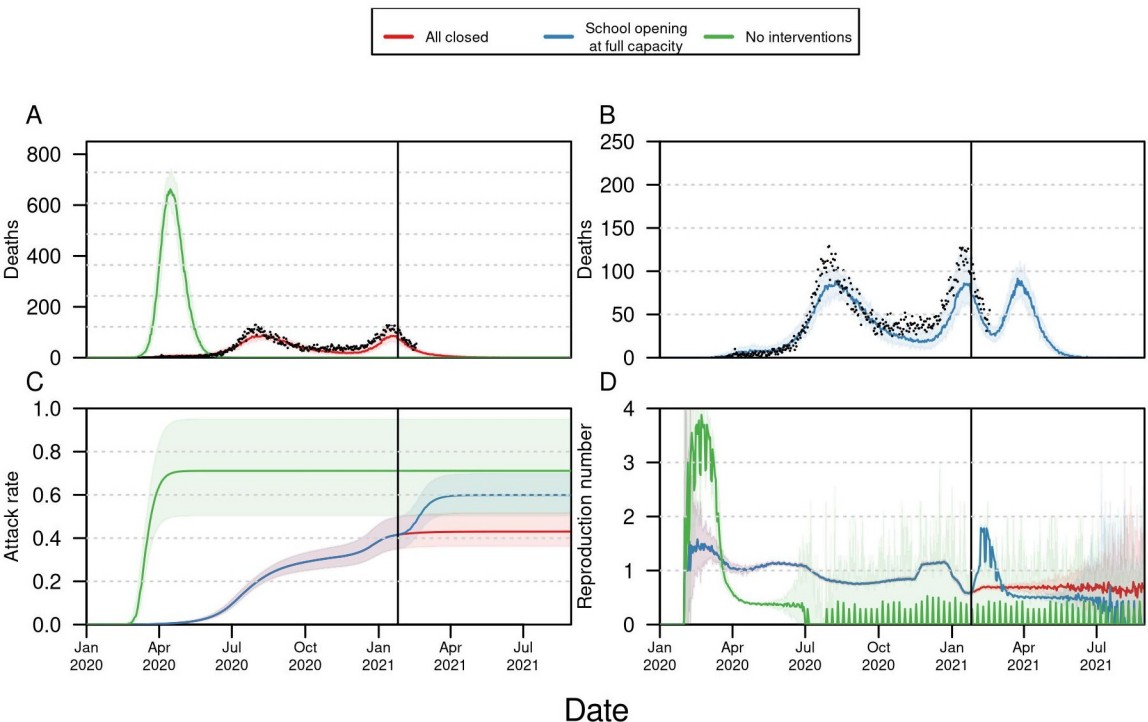

**Fig 2. Projected impact of school reopening in Bogotá, Colombia.** Assumption of lower (50%) susceptibility in <10 years. A) Daily incidence of deaths for two extremes: a scenario in which there were no public health interventions (green), and a scenario with the current public health interventions and assuming schools remain closed for the remainder of the simulation period. B) Daily incidence of deaths in two reopening scenarios: all K-12 schools reopen at full capacity (red). C) Estimated attack rate for the four scenarios considered. D) Estimated reproduction number for the four scenarios considered. All the scenarios were simulated until August 31, 2021.

most of the localities, it underestimated the number of deaths in some localities with older populations, such as Chapinero and Teusaquillo. Overall, the model underestimated the deaths in the older age-group (80+). Another validation point was the infection attack rate, which was estimated as 31.6% (95% CrI:31%-31.8%) by the first week of November in 2020 (Fig 1B), compared to 30% (95%CI: 27%-33%) reported from serological studies during the same period [32]. Our results suggest that this 30% attack rate varied from different regions across the city with the south-west areas having higher attack rates (40%) than the north-east areas (10%-20%) (Fig E in S1 Text).

Based on the assumptions adopted, our model projections show that in the event that schools reopened at full capacity and with no control measures at the end of January, a third wave of COVID-19 could occur, but its impact could be modulated by reducing in-person capacity. Our model estimated a total of 5356 deaths (95% CrI:4951–5690) from February to August 31, 2021, compared to 1906 deaths (95% CrI: 1779–2133) in the event that all schools remained closed (Fig 2). Delaying the date of school reopening reduced the peak of the number of deaths projected within the simulation period for scenarios of high capacity but had a negligible effect on scenarios of low capacity ((Fig 3A and 3D). At full capacity, our projections suggest that reopening on January 25 would have a higher peak of deaths (90 per day) than delaying school reopening to February 25 (78 deaths per day) and March 25 (73 deaths per day). Similar differences were observed at 75% capacity with the highest number of deaths per day (55 deaths per day) reported in the baseline scenario of reopening in January 25, 2021, followed by 48 deaths reopening delayed 1 and 2 months (Fig 3B and 3E). In contrast to the full

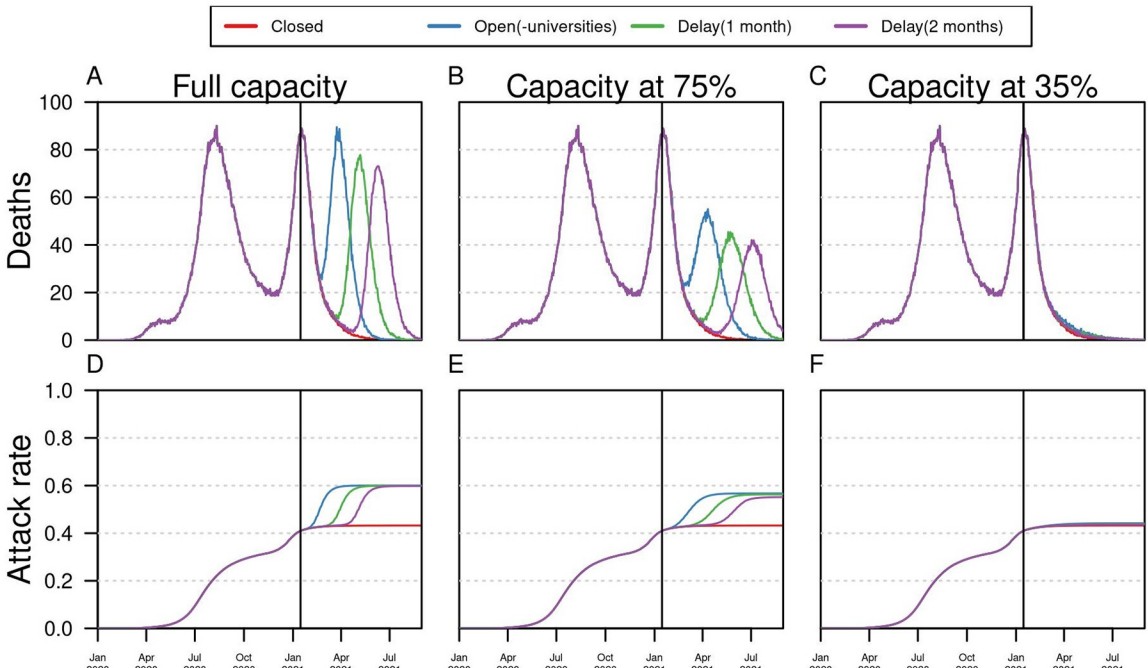

**Fig 3. The impact of delaying school reopening.** Each column shows a different capacity level. Red lines represent a scenario in which all schools remain closed, blue lines represent K-12 schools open, green and purple lines show scenarios of delaying school reopening by 1 and 2 months, respectively. Top panel shows the median estimate of daily incidence of deaths. Bottom panel shows the median estimate of attack rates for each scenario. Vertical black line shows the initial date of school reopening (January 25, 2021). All scenarios were simulated up to August 31, 2021. Assumption of lower (50%) susceptibility in <10 years.

capacity scenario, at 35% the model projections showed that schools alone would not produce a significant increase in the overall number of deaths or the proportion of people infected (Fig 3C and 3F). Although delaying school reopening had an impact in the maximum number of daily deaths, the final percentage of people infected was around 60% for all three dates (Fig 3D), suggesting that the cumulative contribution of school reopening remained the same.

The age of students attending in-person school also affected the projected death toll of COVID-19 in the city. If only children under 6 years of age (pre-K) attended in-person school, a total of 1889 deaths were estimated (95% CrI:1764–2188) at 35% maximum capacity, which was a negligible difference from the baseline scenario of all schools closed. Compared to this baseline scenario, reopening pre-K grades at full capacity resulted in an increase of <200 additional deaths in the whole city (Figs 4A and 5, and Fig M in S1 Text). Scenarios with older students attending in-person school impacted the total number of deaths at different levels depending on the operating capacity. For instance, about 144 additional deaths were estimated when primary school reopened at 50%. In contrast, secondary schools had to operate at a more restricted capacity of 35% to avoid substantially increasing the number of deaths in the city. In fact, at 50% capacity in secondary schools, more than 400 additional deaths were estimated. In the scenario of secondary schools operating at 75% capacity, the model projected a large increase of more than 1600 additional deaths in Bogotá, in comparison to the baseline scenario of schools closed. Furthermore, in the scenario in which students of all ages were able to attend in-person school at some capacity (75% pre-K, 35% primary, 35% secondary), the model projected 431 additional deaths, compared to the closed scenario. At the same level of capacity in pre-K and secondary, but increasing primary capacity to 50%, the number of additional deaths increased to 736. Increasing primary capacity further to 75% resulted in more

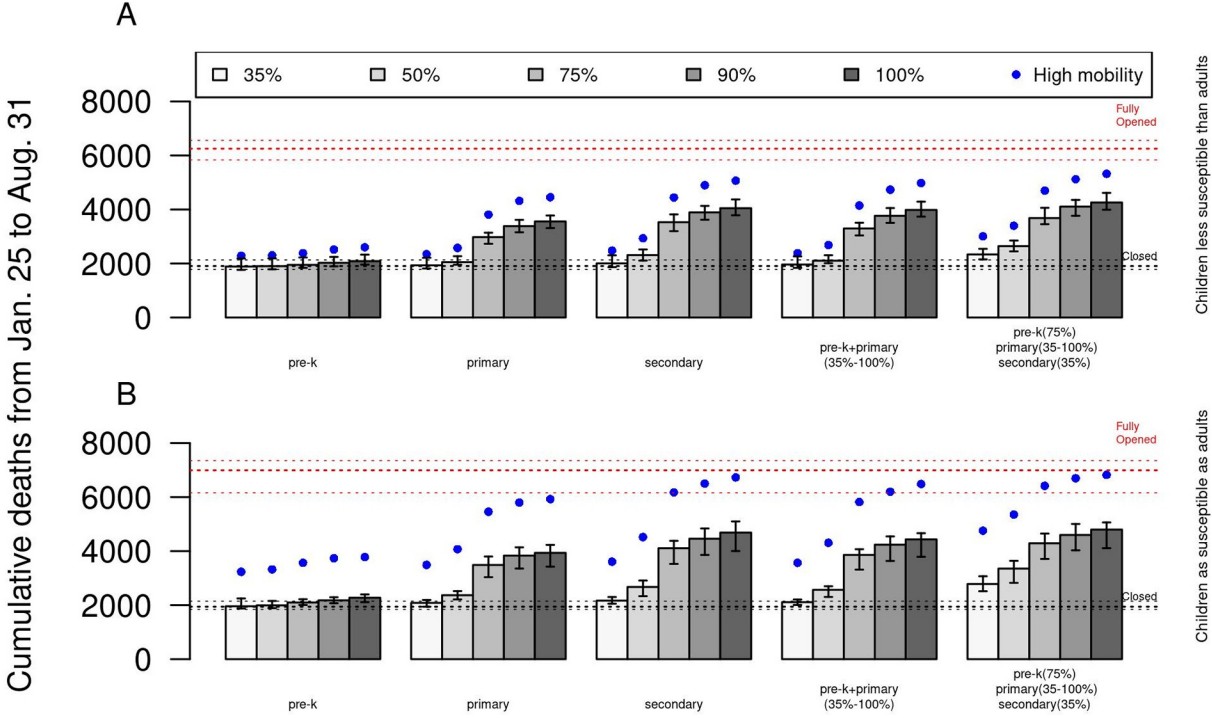

**Fig 4. Total cumulative deaths under different school reopening strategies from January 25 to August 31, 2021.** A) Cumulative deaths of scenarios in which schools reopen by grades with an assumption of lower (50%) susceptibility in <10 years. B) Cumulative deaths of scenarios in which schools reopen by grades with an assumption of equal susceptibility for all ages. From left to right, the first group of bars show exclusive reopening by grade groups in which the other grades remain closed. The fourth group of bars (pre-K+primary) represents a scenario in which pre-K primary and primary reopen at different capacities but secondary remains closed. The last group shows a scenario in which all grades go to in-person school at some level, with pre-K fixed at 75%, secondary fixed at 35%, and primary varying from 35% to 100%. Blue dots show the median estimate of the same scenario with higher mobility in the city when schools reopen. In all scenarios, we assumed long-term protection after SARS-CoV-2 infection.

than 1700 additional deaths. Across all scenarios, the dynamics in time showed that the magnitude of a third wave of infections could have a similar or greater magnitude than the previous two when schools opened at full capacity and no control measures were implemented (Fig 5). Finally, assuming current levels of testing capacity, the positivity of PCR showed an association with the magnitude of future outbreaks (Fig M in S1 Text), which suggested that levels under 10% had a low impact on the city-wide health care system, whereas levels of at 15% or above were correlated to a third wave of large enough magnitude that could put the health system under pressure (Fig M in S1 Text).

Policies of reopening based on the multidimensional poverty index of schools (MPI, high MPI = high poverty in schools) did not show an appreciable difference in the number of deaths (Figs F, I, J in S1 Text). Overall, reopening schools with the highest MPI had a smaller impact on the number of deaths, but differences among schools were small. These results contrast with the impact of COVID-19 being much higher in lower income areas in the south-west of the city (Fig E in S1 Text). At full capacity, these areas might be more insensitive to school reopening given the large proportion of individuals already infected in those areas.

We evaluated our results under alternative assumptions of city-wide mobility, infectiousness and susceptibility to SARS-CoV-2. In the event that school reopening increased the mobility to baseline levels, our results suggest an increase in the impact of reopening at any level under the strategies of reopening by grades or MPI of schools (Fig 4, Fig F in S1 Text).

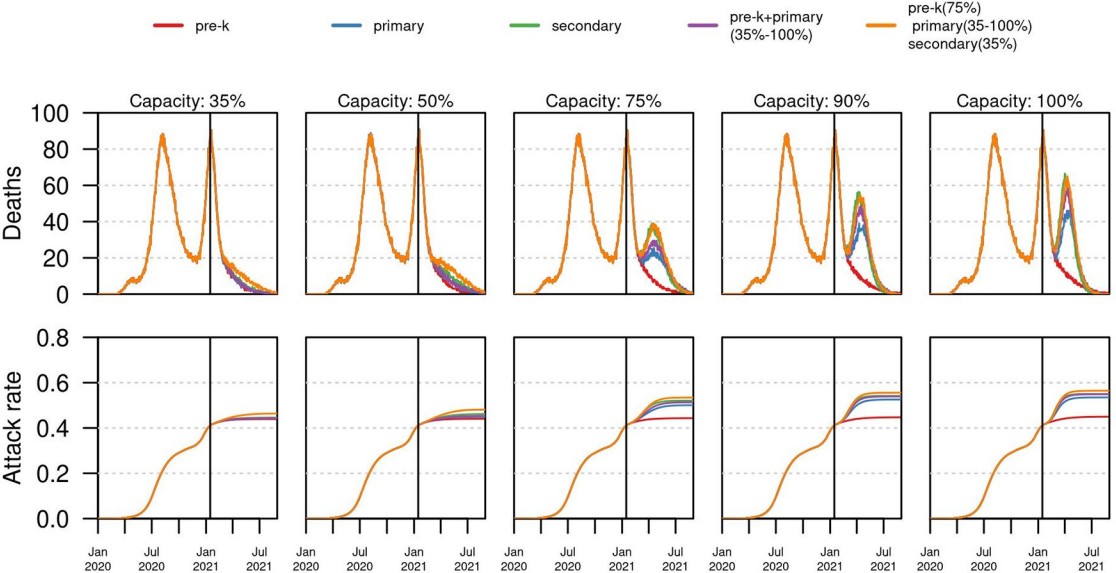

**Fig 5. The impact of school reopening strategies in time.** Each column shows a different capacity level. Top panel shows the median daily incidence of deaths for each reopening strategy based on grades. Bottom panel shows the estimated attack rate for each of the reopening scenarios. Vertical black line shows the timing of school reopening (January 25, 2021). All scenarios were simulated up to August 31, 2021. Assumption of lower (50%) susceptibility in <10 years.

The increase was uniform across all scenarios considered. For instance, reopening pre-K grades increased the number of deaths from 1889 to 2287 at 35% capacity, while pre-K (75%) + primary (35%) + secondary (35%) increased from 2337 to 3008 deaths. Similar increments were observed for the scenarios of reopening by socioeconomic status. In our simulations, school reopening was not the only cause for a third wave in the city. Higher levels of city-wide mobility not linked with schools resulted in an increased death toll at the city level, even when schools remained closed (from 1906 to 2292 deaths). Consequently, the ability of schools to provide continuous in-person teaching could also depend on the overall community levels of mobility. Our simulations showed that it is possible for schools to reopen without a significant increase in the burden of COVID-19 at the city level, but decision makers should evaluate tolerable levels of risk coming from activities in schools and the community. Our results were robust to different assumptions of infectiousness and susceptibility to SARS-CoV-2 infection (Fig K in S1 Text). The impact of reopening strategies based on income and grades remained similar to our main assumption of susceptibility (Fig 4B, and Figs F & N in S1 Text), although the total number of deaths was slightly higher. When schools reopened at full capacity, 4030 additional deaths were estimated with the model, in comparison to 3450 additional deaths with the baseline assumption of susceptibility. In addition, when asymptomatics were assumed to be 75% as infectious as symptomatic individuals [Johannson2021_JAMA], the impact of school reopening was lower (Fig G in S1 Text). Compared to the 2642 deaths estimated under the baseline assumption of infectiousness, at 75% pre-K capacity, 50% primary, and 35% secondary, 2265 total deaths (95%CI: 1989–2910) were estimated.

## Discussion

We evaluated the impact of school reopening strategies in Bogotá during the first semester of 2021, using an agent-based model that includes heterogeneity in transmission, behavior, and adoption of NPIs, which was calibrated to historic trends of COVID-19 in the city. Our

calibration results showed that restrictions in mobility and interactions had an impact in reducing the impact of COVID-19 in the health system. Under an assumption of no public health interventions, we observed a large outbreak with a peak of around 600 deaths per day, and an attack rate close to 75%, which was similar to attack rates observed in unmitigated outbreaks in South America [33]. Furthermore, in these hypothetical scenarios, our model suggests that reduction of control measures could lead to a third outbreak, even with schools closed. Our model projections suggest that school reopening may lead to a substantial increase in SARS-CoV-2 transmission which could lead to a third wave of COVID-19 in Bogotá, Colombia, but this effect can be mitigated by managing the school capacities in older grades, and increasing control measures. These results are consistent with other modeling studies suggesting that younger grades could have a lower impact in transmission than older grades [27, 28, 34].

Our results suggest that reopening schools for in-person instruction at full capacity could result in a third wave of equal or greater magnitude than the first two waves, but the impact on the city-wide dynamics was different depending on the age of students. Particularly, the model showed that reopening pre-K, even at almost full capacity, may not lead to a substantial increase in the overall deaths in the city, given a combination of factors such as lower susceptibility, the total number of students, and the limited contacts of younger children outside of school. The modeling results were insensitive to the assumptions on susceptibility to infection of younger children, suggesting that the reduced effect of lower grades may be caused by the population size and their contact patterns. Previous studies have shown that contact patterns in primary school children are more concentrated in their own grades, as opposed to secondary school children who have more contacts outside their grades [35]. The implications of these results are important for decision makers in public health and the education sector, given that prioritizing the capacity of in-person instruction for younger ages could reduce the risk of a third wave due to school reopening.

Importantly, monitoring the success of these reopening strategies at the school and city-level could be crucial to reduce the risk of a third wave of COVID-19 in the city. Our calibrated model showed that PCR positivity in the whole city had a relationship with transmission. Based on current testing capacity, and after the second wave, at less than 10% of PCR positivity, our results suggest that it may be safe for school reopening with minimal impact in the total number of deaths. In contrast, levels of 10–15% could be indicative of a moderate third wave, and levels greater than 15% could indicate a third wave large enough to put the health system under high pressure.

An important factor for increased transmission of SARS-CoV-2 is the level of mixing in the community. Increased levels of mixing could result in a third wave of COVID-19 in the city, and may have caused the second wave. In fact, we found that the level of mixing needed in the model to reproduce the peak in December related to Christmas and New Year's Eve holidays was greater than any other over the year. These high mixing rates in the community (household to household and family visits) over December resulted in a large and rapid second wave. Similar patterns may be observed over other holidays such as Easter break, but we have not included that assumption in our model. A third wave in Bogotá during the school opening is related to both an increase in mixing patterns within schools and an accompanying increase in community transmission outside school.

The burden of COVID-19 has been heterogeneous across parts of the city, with a larger impact in neighborhoods of lower socioeconomic status. This is not unique to Bogotá. Studies have shown that vulnerable communities are less able to comply with public health interventions that reduce mobility, increasing the burden of COVID-19 in such communities [2]. Our data-driven approach allowed the model to reproduce this geographic heterogeneity,

highlighting the importance of heterogeneity in SARS-CoV-2 transmission, as well as the importance of using models that are capable of reproducing this heterogeneity. Nonetheless, strategies that involved reopening schools based solely on their socioeconomic status were found to have negligible differences in projected burden. This can be explained by a combination of factors. First, the MPI of each school is a metric of the level of poverty of the students attending the school, who do not necessarily live near the school. Instead, students come from different neighborhoods across the city, increasing the probability of infections from high transmission areas being imported in schools located in neighborhoods with low transmission levels. Another factor affecting the small differences in reopening schools by MPI is that students who come from neighborhoods with lower socioeconomic status live in areas with a higher burden of COVID-19, which increases their probability of having been already exposed to the virus. Our results suggest that the risk of reopening schools from different socioeconomic levels is similar but that students from low-income areas may have a higher risk of contracting the virus in their communities due to increased exposure. Nonetheless, these students and their families are impacted the most from the school closures.

Similar to other studies, our results suggest that during the early months of the pandemic in Bogotá, school closures may have contributed to reducing the impact of COVID-19 in the city [17]. The risks of reopening schools should be balanced with the negative societal outcomes of long-term school closures. Our model showed that schools could play a role in a third wave of COVID-19 at high levels of in-person capacity. However, the city-wide impact of school reopening could be greatly reduced by using reduced capacity and having control measures in place. Although in all but the most stringent of cases, we observed an increase in the total deaths, the highest impact of school reopening was found when capacity was high, which resulted in transmission within schools extending to the rest of the community in the city [27].

## Limitations

Our study was set in January 2021 to understand the potential impact of school reopening. Although several other factors have affected the course of COVID-19 in the city, our study focuses on the effect of school transmission in the local context. In total, 16,000 deaths were reported in the period of February 2021—August 2021 with schools partially opened and operating at lower capacity than other activities in the city. Although the magnitude of the third wave was higher than our scenarios, the magnitude of this wave has been attributed to the circulation of the 'mu' variant [36], which was not included in this study.

The evaluation of the impact on COVID-19 dynamics caused by school reopening depends on the epidemiological context. Hence, the predicted effectiveness of interventions to reduce transmission will often depend on whether the intervention reduces the reproduction number below 1, which can be sensitive to the model's parameters [37]. This effect means that, for example, the level at which school reopening capacity is optimized can be difficult to precisely quantify. Our qualitative results should, however, be robust to this effect, and we further mitigate it by exploring a range of scenarios. A caveat to this is that in our calibration, the reproduction number with schools fully opened was substantially greater than 1; if instead, the calibration led to a reproduction number below 1 with schools fully opened, then the impact of school closures would clearly be substantially reduced.

Another limitation of our study is that although our model is a representation of the city including high resolution demographic and geographical data, it is unable to reproduce the full range of heterogeneities in the school system. For instance, we assumed classes are undertaken in classrooms and not outdoors. This could ignore potential benefits of schools with the

capacity to set up outdoor classrooms. Similarly, the model simplifies school structures across socioeconomic status, which in reality may have different characteristics.

Various assumptions were made in our model. Importantly, we assumed that children under 10 years of age are 50% less susceptible than older ages [15]. However, more studies are needed to determine whether children are in fact less susceptible than adults [13]. We also evaluated the impact of school reopening under the assumption of equal susceptibility for children and adults. Even under this assumption, younger grades consistently had a lower impact on transmission than older ones. However, the overall impact of school reopening was slightly higher under the assumption of equal susceptibility. We also assumed that children are able to transmit SARS-CoV-2 at the same level as adults. Although children are less symptomatic than adults [11], published studies suggest that children could be as infectious as adults [13, 38, 39]. We also evaluated a scenario in which relative to symptomatic infections, asymptomatic infectiousness was slightly lower (75%). Under this assumption there was a reduced impact of school reopening. This reduction was proportionally larger for scenarios of low or moderate capacity, but at higher capacities the reduction was lower. The ability of children to transmit the virus emphasizes the importance of face-mask adherence, maintaining physical distancing in schools, and other interventions, such as controlling capacities in schools.

Another assumption made in the model is that levels of mobility would increase up to levels seen in November, 2020. However, the model does not include adaptive behaviors, such as parents changing schedules in the case that their children attend in-person school, which could have an impact on mobility and contacts across the city. Mobility could also increase by students using public transportation to go to school, which was not included in the model. Hence, mobility could increase even more than levels seen in November 2020. Consequently, we assumed a scenario with higher mobility up to baseline pre-pandemic levels. At this level of mobility, deaths increased slightly and uniformly across all scenarios studied. Although we are unable to project the full extent of future mobility and levels of contacts within the city, this result highlights the importance of continuing control measures in the city to maintain acceptable levels of transmission when schools reopen.

We considered a reduced set of possible reopening strategies to focus on quantifying the impact of school capacity by age and socioeconomic status. Another strategic aspect not considered is the effect of face-mask adherence within school, which has been explored in similar analyses of school reopening [30]. Instead, we set the baseline level of face-mask adherence to 75%, based on city surveys. Furthermore, we did not consider reactive interventions to control the spread of SARS-CoV-2 within schools, such as contact tracing, classroom closures, or individual school closures. Another simplification of the school reopening strategy is that we simulated uniform mandates and compliance with public health measures across the city. The reality is that some schools would be able to enforce interventions more than others. Nonetheless, our simulations represent an average of the city-wide reopening strategy. In general, our results highlight the importance of controlling school capacity at different levels depending on the school grades.

Another limitation of the model is the quality of the data used to calibrate the model. We focused on daily number of deaths because death reports are more reliable than case data. Nonetheless, the number of deaths in the city can also be underreported as it has been estimated in other countries [40]. To increase the reliability of our model calibration, we validated the model to other data types not included in the calibration, such as the infection attack rate. We used this calibrated model of COVID-19 in Bogotá to evaluate scenarios of school reopening, but our results do not represent predictions of the future course of the epidemic in the city. Instead of predicting the course of the epidemic, we used a large-scale agent-based model of SARS-CoV-2 transmission that incorporates multiple data types to better understand the

potential impact of schools in the COVID-19 dynamics in the city under different hypothetical strategies of school reopening. The reopening strategies evaluated in this study does not include reactive measures that schools could take to reduce the impact of outbreaks once they are identified. This means that our results could underestimate the impact of school reopening in some aspects and overestimate it in others. Although in the school opening scenarios we have assumed the current mobility levels will increase up to November levels and a scenario of high mobility with baseline levels of mobility, the model is unable to estimate the levels of contacts outside schools increased for other reasons and what would be the impact on intra-school transmission.

The model results strongly depend on the quality of the synthetic population incorporated in the model. A limitation of the model is that our synthetic population does not incorporate all potential group quarters where populations at risk could live, such as informal nursing homes, or monasteries. Incorporating additional sources of data to inform the synthetic population could improve the model's ability to reproduce the dynamics of COVID-19 in localities where it currently underestimates its impact. Furthermore, the overall structure of the synthetic population underestimates the population under 20 years of age. This implies that our model simulations could underestimate the number of infections in this group age in the city. Although, a younger population would result in a lower overall fatality rate due to COVID-19.

The model does not explicitly include the potential impact of public transportation or school's transportation. Finally, the model does not include potential impact of waning immunity or other variants with increased transmission or immunity escape capacities, and does not include potential vaccination scenarios.

## Methods

### Data

Demographic data was obtained from IPUMS-International, and the city planning secretary of Bogotá [41, 42]. Demographic data on long-term care facilities were obtained from the Census and the ministry of health [43, 44]. We manually geo-located these institutions using google maps.

Information about the number of schools, their capacity by age, and geo-location were obtained from the city's Secretary of Education, which also provided us with a list of the Multidimensional Poverty Index (MPI) for each school. The MPI of each school represented the level of poverty of its students, not the location of the school. For institutions of superior education, we obtained a list with capacities from the national Ministry of Education [45] and manually geo-located them using google maps. We obtained data-sets for workplaces, including the number of workers and geo-location of each formal and informal workplace in the city, from the Secretary of Education.

We used publicly available data to approximate trends in the adoption of public-health interventions, such as lockdowns and the use of face masks. For lockdowns, we used the Google Mobility Reports [46] on the time-varying proportional change of people staying at home since March, 2020. We later adjusted the magnitude of this time-series to fit the model. To approximate the geographical variation of lockdown compliance, we combined the time-varying trends from Google Mobility Reports with data from the *Grandata* project [47], which includes changes in mobility by day at the census-tract level (Unidad Catastral) but were not as frequently updated as the reports from Google. The adoption of face masks was approximated using data from google trends on the specific search terms 'tapabocas' and 'mascarilla' from February until October, 2020 [48]. Assuming that people who bought masks would subsequently wear them, we computed the cumulative interest in those terms and used a scaling

factor in the calibration step to estimate the proportion of people wearing face masks over time.

We used daily incidence data on deaths from the surveillance system of the National Institute of Health (INS) [49]. We also used data stratified by age and locality in Bogotá from the city's Secretary of Health, to validate the model performance. Serological studies were also used to compare model performance [32].

### Description of agent-based model

We modeled the dynamics of SARS-CoV-2 transmission with an agent-based model using a modified version of the platform FRED [50], which was originally developed to simulate influenza pandemics at the University of Pittsburgh. This version of the model has been described elsewhere [30]. This model has also been used previously to simulate COVID-19 dynamics in school reopening in Indiana [30] and to forecast the weekly incidence of death in seven states in the United States as well as to study the impact of non-pharmaceutical interventions [51, 52]. In our model, each inhabitant of Bogotá is modeled as an agent who has a set of daily activities, such as school attendance or commuting to work (Fig A in S1 Text). Transmission of the pathogen can occur when an infectious person visits the same place a susceptible person visited the same day. We assumed that proportion of the overall infectious people in the city would visit long-term care facilities, potentially infecting their residents. Finally, the probability of transmission partly depends on the number of effective contacts that a person has for each location type. These numbers of contacts were assumed to be those previously calibrated values to influenza for each location type [50].

Transmission and disease progression is based on a modified SEIR model. Latency and infectious periods were drawn from distribution calibrated to the average generation interval in Singapore [53]. The probability of developing symptoms increases with age [10]. Similarly, the probability of death increases with the age [54]. We assume that agents who recover from infection acquire long-term immunity. We assumed children and adults have the same capacity to transmit the virus to others upon exposure, although they were less likely to develop symptoms. We assumed that asymptomatic and symptomatic infectious individuals had a similar probability of infecting a susceptible agent upon exposure, but relaxed this assumption in an alternative analysis in which asymptomatic infectiousness was set to 75% that of symptomatic infections [55]. Based on limited evidence on children susceptibility, we assumed two possibilities i) that children under 10 years of age were 50% less susceptible to infection compared to older children and adults ii) that children have the same susceptibility to infection as adults [14].

Non-pharmaceutical interventions were incorporated in the model to modify agents' behavior to curb the burden of COVID-19. We simulated lockdowns by restricting agents' mobility to their household and local community based on daily reports of human mobility in the city [46]. The effect of people wearing face masks was included in the model by reducing the probability of transmission of an susceptible individual upon exposure. The efficacy of this measure was determined as the lower bound of the odds ratio from estimates of SARS-CoV efficacy in non-health care settings (aOR: 0.73) [56]. The temporal trends of people wearing face masks was adjusted from google trends on specific search of face masks in Bogotá ('tapabocas,' 'mascarilla') [48]. The proportion of people wearing face masks depended on the specific location and the age of the agent. Only people older than 7 were eligible to wear a face mask. For workplace and community, temporal trends from google trends were adjusted with a scaling factor in the calibration step. We assumed that people did not wear face masks in their households. In the event that schools reopen, we assumed that 75% of students older than 7 years of age would properly wear face masks.

The model includes schools that represent the set of private and public schools in Bogotá in terms of age, enrollment, location, and size. Transmission of the virus in schools can occur because of contacts inside the classroom or with the rest of the school [50]. We assumed that for a person in the school, the number of contacts in the classroom is double the number of contacts with the rest of the school. The size of each classroom was determined by age in agreement with the average size by grade in the city schools. The model also includes the population of teachers.

## Synthetic population

We created a synthetic population that matches geographical and demographic characteristics of the population in Bogotá. We used publicly available micro-data from the IPUMS-International database [41] and used an iterative proportional fitting algorithm using the simPop package in R to fit age, household-composition, and population size by each census tract unit (Unidad Catastral) [57]. We also included long-term care facilities in the model based on data from the ministry of health. The synthetic population was fit to census-tract data and it also represents the city-wide population by age and household population (Fig H in S1 Text). The geographical density Bogotá is distributed in neighborhoods and localities, which contain several neighborhoods. The population density by census tract is shown in (Fig H in S1 Text). Also, the precise location of households, schools, and workplaces is shown in (Fig H in S1 Text). We focused on the urban localities and omitted the locality of Usme, which is mainly rural.

In the synthetic population, students in pre-K, primary, and secondary school were assigned to school based on data from the Secretary of Education for each grade. Students were assigned to a school in three sequential steps. First, for each student, a list of schools with availability for the student's age was created. Then, we used data from the Secretary of Education to determine a matrix of locality of residence vs locality of school. Based on this matrix, we selected a locality to assign the student's school. Third, we assigned the school of the student based on two criteria, if the locality is the same as the student's household, we assign the student to the closest school with availability, if the locality is not the student's household locality, we assigned the school at random within that locality. For students in higher education, such as universities, we obtained a list of institutions with their student capacity from the Ministry of Education [45]. We randomly assigned students in higher education institutions based on their capacity.

Workers were assigned to workplaces based on a data set of formal and informal workplaces. This database included the number of workers and geo-location of the workplace. We used a mobility survey in Bogotá to create a matrix of locality of household vs locality of workplace. Based on this matrix, we assigned workers to workplaces based on distance and capacity.

## Model initialization and calibration

To reproduce the timing of SARS-CoV-2 importation in Bogotá, we initialized the model based on international and domestic importations in the city using case fatality risk and locally reported death data. Detailed description of these methods are described elsewhere [30, 58]. We fitted a GAM to the mobility trends from the percentage change on mobility for places of residence, and assumed that future mobility would increase up to values observed in November, 2020. We defined the maximum mobility in the city as 0% of people sheltering in place and the minimum mobility in the reports as 100% of people sheltering in place. Then, we scaled these trends based on a scaling factor that we calibrate. We adjusted the numerical

values of six model parameters to reproduce the daily incidence of deaths in Bogotá. Namely, the scaling factor for imported infections, a scaling factor for importation of infections to long-term care facilities, the probability of transmission upon exposure, the adherence with shelter-in-place and face-mask recommendations, and a percentage increase of community contact during the holidays. We calculated the likelihood of the model given the observed daily incidence of deaths for 2,000 simulations of the model with combinations of these parameters, $\vec{\theta}$, using a sobol design sampling algorithm with the sobolDesign function in R [59, 60]. We then sampled from these 2,000 parameter sets based on their likelihood, which was calculated as $L\left(\vec{\theta} \vee D_t\right) = \text{Negative Binomial}(r, p)$, where $D_t$ is the daily incidence of death on day $t$ and $r$ and $p$ are size and probability parameters, respectively. We informed $r$ and $p$ using the conjugate prior relationship between a beta prior and negative binomial likelihood.

We validated the model with data excluded from the calibration process. Serological studies were carried out in Bogotá between October 26th and Novebmer 17th, 2020 to estimate the proportion of the population infected with SARS-CoV-2 [32]. We estimated daily attack rate in our model and compared the values to the serological study.

We also contrasted our model to the daily positive rate of PCR and antigen tests. We assumed perfect specificity and sensitivity of 0.85 for PCR [61] and 0.75 for antigen tests [62]. The proportion of positive tests were calculated as

$$P(P \vee T) = \text{sensitivity}(P(C \vee T) + P(I \vee T)) + (1 - \text{specificity})P(U \vee T),$$

where T refers to PCR or antigen tests administered, C to symptomatic infections, I to asymptomatic or pre-symptomatic infections, and U to uninfected individuals. As explained elsewhere [30], P(C|T) can be expressed as

$$P(C \vee T) = \frac{P(C)}{P(C) + r(1 - P(C))},$$

where $r = P(T \vee \neg C)/P(T \vee C)$. P(I) and P(U) can be written as

$$P(I \vee T) = \frac{rP(I)}{P(C) + r(1 - P(C))}$$

and

$$P(U \vee T) = \frac{rP(U)}{P(C) + r(1 - P(C))}.$$

## School reopening scenarios

We simulated different school reopening scenarios with the aim of evaluating the impact on COVID-19 dynamics in the city at different levels of in-person school attendance. To inform public policy, we based our scenarios on discussions with the Secretary of Education of Bogotá. We focused on different attendance levels with different priorities based on age. Also, we simulated scenarios in which young children had priority of in-person attendance to provide scenarios in which single mothers with young children could go to work. Similarly, we designed scenarios in which we incremented the in-person attendance of older students, given that these students could be at risk of unemployment and poverty. We also focused on reopening strategies based on the MPI and geographical location of schools.

We simulated reopening strategies of grades including, pre-K, primary, and secondary school. We modeled varying degrees of school capacity by modulating the probability of a student to go to school on a specific day. In the model, reduced capacity does not imply greater physical distancing within schools. We varied the capacity of reopening for in-person students from 35% to 100% for each set of grades. We also evaluated the impact of reopening pre-K and primary schools together at similar capacity levels from 35% to 100%. Finally, we simulated a scenario in which students from all ages were able to attend in-person school at some level with 100% pre-K, 50% primary school, and secondary school capacity varying from 35% to 100%.

The multidimensional poverty index (MPI) is an international measure of poverty that includes monetary poverty metrics and other acute deprivations in health and living standards [63]. We used an adjusted MPI for each school, which represents the overall intensity of poverty in the school's students. Then, we sorted schools based on their MPI and student population size, and grouped the schools based on their population quartile in four groups (MPI Q0-Q1, Q1-Q2, Q3-Q4) from lower to higher MPI index. To estimate the effect of MPI of schools in school reopening, we simulated exclusive reopening for each of the four determined groups.

We also simulated extreme scenarios of school reopening in which schools remain at their current level of attendance or they are open at full capacity. Finally, we evaluated the impact of delaying school reopening by 1 or 2 months from the initially planned reopening (January 25, 2021). In all simulations, we evaluated the impact of reopening schools as the total number of deaths reported in the city from January 25 to August 31, 2021.

## Supporting information

**S1 Text. Additional modeling results.**
(DOCX)

## Author Contributions

**Conceptualization:** Guido España, Zulma M. Cucunubá, Hernando Diaz, Sean Cavany.

**Data curation:** Guido España, Hernando Diaz, Nelson Castañeda, Laura Rodriguez.

**Formal analysis:** Guido España, Zulma M. Cucunubá, Hernando Diaz.

**Investigation:** Guido España, Zulma M. Cucunubá.

**Methodology:** Guido España, Zulma M. Cucunubá, Hernando Diaz, Sean Cavany, Nelson Castañeda, Laura Rodriguez.

**Supervision:** Zulma M. Cucunubá.

**Validation:** Guido España, Zulma M. Cucunubá.

**Visualization:** Guido España.

**Writing – original draft:** Guido España, Zulma M. Cucunubá, Hernando Diaz, Sean Cavany, Nelson Castañeda, Laura Rodriguez.

**Writing – review & editing:** Guido España, Zulma M. Cucunubá, Hernando Diaz, Sean Cavany, Nelson Castañeda, Laura Rodriguez.

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
