## [Decision Letter · Decision Letter 0]

3 Nov 2021

PGPH-D-21-00520

Optimization of school reopening on the impact of the COVID-19 third wave in Bogotá, Colombia

Dear Dr. España,

Thank you for submitting your manuscript to PLOS Global Public Health. After careful consideration, we feel that it has merit but does not fully meet PLOS Global Public Health’s publication criteria as it currently stands. Therefore, we invite you to submit a revised version of the manuscript that addresses the points raised during the review process.

We look forward to receiving your revised manuscript.

Kind regards,

Paolo Angelo Cortesi, PhD

Academic Editor

Journal Requirements:

1. Please update the completed 'Competing Interests' statement, including any COIs declared by your co-authors. If you have no competing interests to declare, please state "The authors have declared that no competing interests exist". Otherwise please declare all competing interests beginning with the statement "I have read the journal's policy and the authors of this manuscript have the following competing interests:"

2. Please provide separate figure files in .tif or .eps format only, and remove any figures embedded in your manuscript file.  If you are using LaTeX, you do not need to remove embedded figures.

3. We have noticed that you have uploaded supporting information but you have not included a list of legends.  Please add a full list of legends for all supporting information files (including figures, table and data files) after the references list. 

4. State what role the funders took in the study. If the funders had no role in your study, please state: “The funders had no role in study design, data collection and analysis, decision to publish, or preparation of the manuscript.”

5. Please ensure that the funders and grant numbers match between the Financial Disclosure field and the Funding Information tab in your submission form. Note that the funders must be provided in the same order in both places as well. Currently, not all funders are include the Funding Information section.

Additional Editor Comments (if provided):

Reviewers' comments:

Reviewer's Responses to Questions

**Comments to the Author**

1. Does this manuscript meet PLOS Global Public Health’s publication criteria? Is the manuscript technically sound, and do the data support the conclusions? The manuscript must describe methodologically and ethically rigorous research with conclusions that are appropriately drawn based on the data presented.

Reviewer #1: Yes

Reviewer #2: Yes

2. Has the statistical analysis been performed appropriately and rigorously?

Reviewer #1: Yes

Reviewer #2: Yes

3. Have the authors made all data underlying the findings in their manuscript fully available (please refer to the Data Availability Statement at the start of the manuscript PDF file)?

Reviewer #1: Yes

Reviewer #2: Yes

4. Is the manuscript presented in an intelligible fashion and written in standard English?

Reviewer #1: Yes

Reviewer #2: No

5. Review Comments to the Author

Reviewer #1: The article is a compelling study of SARS-CoV-2 in Bogota, Colombia using an agent-based model.

Results presented, which are based on and validated by appropriate datasets, demonstrate potential effects of different school reopening scenarios.

In addition to being topical as children across the world returned to school throughout 2021, this study's impact extends beyond the SARS-CoV-2 pandemic after different policies have played out.

The overall methodology undertaken appears reasonable and sound.

The authors have done a nice job addressing some uncertainties about SARS-CoV-2 infections, specifically susceptibility in children and young adults, that are pertinent to simulations under any mathematical model.

However, I believe the manuscript would benefit from some revisions.

I urge the authors to consider the following comments, questions, and recommendations:

#### Major comments

1. Agent-based models can be notoriously difficult to understand. It would be helpful to readers if the authors could provide a high-level summary of the model. I agree that referring interested readers to Guido et al 2020 is appropriate given its excellent account of technical details, but a simple schematic may help the article reach a wider audience. In particular, the summary should help one understand how various data sources feed into the model and where key assumptions may affect model outputs (e.g. deaths and attack rates).

2. The authors give a proper introduction as to the relevance of school reopening and its impact to public health in Bogota, Colombia. However, the rationale for the methodology is not addressed. For example, what is the benefit of using this stochastic agent-based model vs a simpler stochastic/deterministic compartmental model? Adding 1-2 sentences somewhere in the paragraph beginning at Line 81 would suffice.

3. The authors explore a wide range of scenarios for reopening but do not discuss why these scenarios (e.g. opening schools only for younger children) were chosen. Given the nature of the journal, I feel it is appropriate for the authors to expand why these scenarios are pertinent to Bogota and whether they relate to any public health policies discussed/implemented in the Colombia as a whole.

4. In Figure S2, model estimates of deaths closely align with data in most localities. In particular, the sudden uptick around Jun/July 2020 is recapitulated. However, two localities stick out, Chapinero and Teusaquillo, where (i) quantitatively, estimates and data diverge in terms of magnitude in deaths, and (ii) qualitatively, the surge never occurs in the median trajectories. Yet, the 95% CrIs do capture the shape of the data. Can the authors comment on why the model does not predict a median trajectory of similar shape to the data? Interestingly, proportions of deaths (Figure S3) are underestimated in older populations (80-120) in both localities pointed out, as well as others.

5. Based on Figure S5, MPI and deaths do not seem to be correlated. That is, no MPI subgroup seems to be dispproportianately burdened by deaths in simulations across scenarios and assumptions about susceptibility. Are there any model outputs that seem affected by MPI? If not, what role does MPI play in the simulation? This is alluded to in the discussion, but not quantified. Is it possible to illustrate how these subgroups are distributed over geography?

6. The important conclusion at Lines 204-206 is a little difficult to understand. The level of community contact (based on mobility) increases deaths across all scenarios, but what exactly constitutes an acceptable burden of cases? That is a difficult philosophical question that modelling alone cannot answer, and it is not clear how it *should* be addressed. If it is not too difficult, the authors may be able to examine how the additional deaths due to reopening are distributed across localities and schools. One could argue that a distribution where deaths concentrate in a locality or around a particular school indicates an unfavorable/unjust/riskier scenario for a society compared to a more randomly distributed pattern.

7. The effect over time of reopening at or near full capacity is quite stark, even when only primary schools reopen and at a lower susceptibility in children. Can the authors confirm that this effect in simulations is strictly due to reopening? Notably, the population pyramid in Figure S7A indicates that the synthetic population has fewer children/young adults under 20 compared to the census data. This is not necessary for the main text, but it would be informative to somehow quantify the model's sensitivity to the size of this demographic in this specific scenario. Weak or lack of sensitivity would suggest that the contact patterns in these groups are driving the surge.

8. Finally, the authors should spend some time improving the prose of the text to improve its flow and clarity.

#### Minor comments

1. The caption on Figure S1 states blue is used to indicate lower mobility but the figure legend uses blue for higher mobility.

2. In the main text there are a few references to Figure S5 where I believe the intended reference is Figure S6.

3. Figure S8 demonstrates a third surge due to increase capacity in reopening schools. It would be nice to zoom in on one of these examples to be able to see disparities in the different MPI groups since they are completely overlapping. A reference in the text would also suffice.

4. At line 195, I would subsititute "insensitive" for "resilient" because the observed effect is not an innate property of areas with vulnerable populations.

Reviewer #2: The topic is extremely interesting, as the issue of school opening has been, and will likely be, one of the main areas of decision making during the Covid pandemic.

The study is well planned, and carried out with appropriate tools and detailed analysis.

The outcomes of the study show a variety of responses to school closures or reopening, with the possibility of taking measures so that school reopening has a very mild effect on the course of the pandemic.

There are three issues with the current manuscript, though, that should be addressed in a (minor) review.

1(a) Although compartmental mathematical models have been an essential tool in analyzing and forecasting pandemic trajectories, one should be aware of the limitations.

A general discussion is, for instance, in Roda et al.: Why is it difficult to accurately predict the COVID-19 epidemic?, Infect Dis Model. 2020; 5: 271–281.

For the specific case of schools one relevant analysis is in Gandolfi et al. "A new threshold reveals the uncertainty about the effect of school opening on diffusion of Covid-19." arXiv preprint arXiv:2104.04136 (2021).

These issues should be addressed by the authors, at the very least in the "Limitations" section, but possibly relating to the results of the paper to the above references in other sections.

1(b) The research relies on mortality data, so, in addition to the instabilities mentioned in 1(a), one has also an issue of data reliability. For instance, Karlinsky et al.: Tracking excess mortality across countries during the COVID-19 pandemic with the World Mortality Dataset, eLife 2021;10:e69336, indicates that mortality was underestimated in Bolivia by a factor of 1.2. This should also be addressed, at least in the "Limitations" section.

2. Various parts a not well structured and not so easy to follow.

The section on Results could be more focused. I understand there is a variety of factors affecting the outcome of school reopening, but perhaps the authors should try to find a common thread though the various scenarios. This could possibly be centered around Figures 4 and 5, for instance.

The Discussion section is even less structured. Especially from Line 273 to Line 348, the presentation is a stream of short remarks on modeling choices and possible shortcomings that ends up becoming a little confusing. Some of the statements do actually refer to limitations more than discussion. Perhaps this part could be streamlined by moving some remarks to limitations, and adding perhaps a table summarizing the selected parameters.

3. There are a number of typos. The overall paper should be revised in this sense. Here is a short list.

Line 25: perhaps it should be "priority in fighting poverty"

Line 273: transmision - > transmission

Line 289: christmas -> Christmas

Line 294: the sentence "A third wave in Bogotá, related to school opening is related to both, an increase in mixing patterns within schools and an accompanying increase in community transmission outside school" is not clear.

Line 304: " level of poverty of schools attending the school" ??

Line 306: being -> having been

Line 307: the sentence "Another factor associated with the small difference across MPI values is that schools attract few students from various neighborhoods around the city, increasing the probability that schools located in neighborhoods with low transmission could have imported infections from neighborhoods with high transmission" is not clear.

Line 312: However -> However,

Line 316: "although" does not make sense here.

Lines 421, 422: increases vs. increased (there is no time coordination)

Line 438: (aOR: 0.73) ??

Line 464 and others: prek -> pre-K

6. PLOS authors have the option to publish the peer review history of their article (what does this mean?). If published, this will include your full peer review and any attached files.

**Do you want your identity to be public for this peer review?** For information about this choice, including consent withdrawal, please see our Privacy Policy.

Reviewer #1: No

Reviewer #2: No

---

## [Decision Letter · Decision Letter 1]

1 Mar 2022

PGPH-D-21-00520R1

Optimization of school reopening on the impact of the COVID-19 third wave in Bogotá, Colombia

Dear Dr. España,

Thank you for submitting your manuscript to PLOS Global Public Health. After careful consideration, we feel that it has merit but does not fully meet PLOS Global Public Health’s publication criteria as it currently stands. Therefore, we invite you to submit a revised version of the manuscript that addresses the points raised during the review process.

We look forward to receiving your revised manuscript.

Kind regards,

Paolo Angelo Cortesi, PhD

Academic Editor

Journal Requirements:

1. Please provide us with a direct link to the base layer of the map used in Figure S2, Figure S5, and Figure S8, and ensure this location is also included in the figure legend. 

Please note that, because all PLOS articles are published under a CC BY license (creativecommons.org/licenses/by/4.0/), we cannot publish proprietary maps such as Google Maps, Mapquest or other copyrighted maps. If your map was obtained from a copyrighted source please amend the figure so that the base map used is from an openly available source.

Please note that only the following CC BY licences are compatible with PLOS licence: CC BY 4.0, CC BY 2.0  and CC BY 3.0, meanwhile such licences as CC BY-ND 3.0 and others are not compatible due to additional restrictions. If you are unsure whether you can use a map or not, please do reach out and we will be able to help you. 

The following websites are good examples of where you can source open access or public domain maps:

Additional Editor Comments (if provided):

Before to accept the paper you have to address the last minor revisions suggested by one reviewer:

- sure, the models are agent based: the word "compartmental" was a typo in the report. Still, I think that some of my comment there was appropriate, especially the part about the instability in the effect of school policies which seems to appear in the results presented in this paper.

- I might have missed it, but I did not find a summary of what actually happened in the period February-August 31, 2021: were schools closed? What was the observed mortality?

- Years are not shown in the figures. I guess Figure 1 is for 2020 and Figure 2 and 4 are for 2020 and part of 2021.

Reviewers' comments:

Reviewer's Responses to Questions

**Comments to the Author**

1. If the authors have adequately addressed your comments raised in a previous round of review and you feel that this manuscript is now acceptable for publication, you may indicate that here to bypass the “Comments to the Author” section, enter your conflict of interest statement in the “Confidential to Editor” section, and submit your "Accept" recommendation.

Reviewer #1: All comments have been addressed

Reviewer #2: (No Response)

2. Does this manuscript meet PLOS Global Public Health’s publication criteria? Is the manuscript technically sound, and do the data support the conclusions? The manuscript must describe methodologically and ethically rigorous research with conclusions that are appropriately drawn based on the data presented.

Reviewer #1: Yes

Reviewer #2: Yes

3. Has the statistical analysis been performed appropriately and rigorously?

Reviewer #1: Yes

Reviewer #2: Yes

4. Have the authors made all data underlying the findings in their manuscript fully available (please refer to the Data Availability Statement at the start of the manuscript PDF file)?

Reviewer #1: Yes

Reviewer #2: Yes

5. Is the manuscript presented in an intelligible fashion and written in standard English?

Reviewer #1: Yes

Reviewer #2: Yes

6. Review Comments to the Author

Reviewer #1: The authors have addressed all comments and polished the manuscript. I fully recommend acceptance for publication.

Reviewer #2: The authors took care of all the points raised by the referees, and I think the results are now well presented. I still have two comments for the authors:

- sure, the models are agent based: the word "compartmental" was a typo in the report. Still, I think that some of my comment there was appropriate, especially the part about the instability in the effect of school policies which seems to appear in the results presented in this paper.

- I might have missed it, but I did not find a summary of what actually happened in the period February-August 31, 2021: were schools closed? What was the observed mortality?

- Years are not shown in the figures. I guess Figure 1 is for 2020 and Figure 2 and 4 are for 2020 and part of 2021.

7. PLOS authors have the option to publish the peer review history of their article (what does this mean?). If published, this will include your full peer review and any attached files.

**Do you want your identity to be public for this peer review?** For information about this choice, including consent withdrawal, please see our Privacy Policy.

Reviewer #1: No

Reviewer #2: No

---

## [Editor Report · Decision Letter 2]

17 May 2022

Designing school reopening in the COVID-19 pre-vaccination period in Bogotá, Colombia: a modeling study

PGPH-D-21-00520R2

Dear Mr. España,

We are pleased to inform you that your manuscript 'Designing school reopening in the COVID-19 pre-vaccination period in Bogotá, Colombia: a modeling study' has been provisionally accepted for publication in PLOS Global Public Health.

Best regards,

Paolo Angelo Cortesi, PhD

Academic Editor